# Development of clinical prediction rule for the requirement of endoscopic papillary large balloon dilation (EPLBD) on endoscopic CBD stone clearance

Chote Wongkanong[1], Jayanton Patumanond[2], Thawee Ratanachu-ek[3], Sunhawit Junrungsee[4,5], Apichat Tantraworasin[2,4,5]*

1 Department of Surgery, Pattani Hospital, Pattani, Thailand, 2 Center for Clinical Epidemiology & Clinical Statistics, Faculty of Medicine, Chiang Mai University, Chiang Mai, Thailand, 3 Surgical Endoscopy Unit, Department of Surgery Rajavithi Hospital, Department of Medical Services, Ministry of Public Health, Mueang Nonthaburi, Thailand, 4 Clinical Surgical Research Center, Chiang Mai University, Chiang Mai, Thailand, 5 Department of Surgery, Faculty of Medicine, Chiang Mai University, Chiang Mai, Thailand

* Apichat.t@cmu.ac.th

**Data Availability Statement:** All relevant data are within the paper and its.

## Abstract

### Introduction

To develop a simplified scoring system for clinical prediction of difficulty in CBD stone removal to assist endoscopists working in resource-limited settings in deciding whether to proceed with an intervention or refer patients to a center capable of performing additional procedures and interventions.

### Methods

This study included patients with CBD stones who underwent ERCP at Pattani Hospital between August 2017 and December 2021. Retrospective cohort data was collected and patients were categorized into two groups: bile duct stones successfully treated by endoscopic biliary sphincterotomy and extraction compared to the former method combined with EPLBD. We explored potential predictors using multivariable logistic regression. The chosen logistic coefficients were transformed into a scoring system based on risk with internal validation via bootstrapping procedure.

### Results

Among the 155 patients who had successful endoscopic therapy for bile duct stones, there were 79 (50.97%) cases of endoscopic biliary sphincterotomy, EPLBD and extraction versus 76 (49.03%) cases without EPLBD. The factors used to derive a scoring system included the size of CBD stones >15 mm, the difference between the stone and distal CBD diameter >2mm, distal CBD arm length <36 mm and stone shape. The score-based model's area under ROC was 0.88 (95% CI: 0.83, 0.93). For clinical use, the range of scores from 0 to 16, was divided into two subcategories based on CBD stone removal difficulty requiring EPLBD to derive the PPV. For scores <5 and ≥ 5, the PPV was 23.40 (p <0.001) and 93.44

**Funding:** The funders had no role in study design, data collection and analysis, decision to publish, or preparation of the manuscript.

**Competing interests:** The authors have declared that no competing interests exist.

(p <0.001) respectively. The Bootstrap sampling method indicated a prediction ability of 0.88 (AuROC, 95% CI: 0.83, 0.94).

## Conclusion

This scoring system has acceptable prediction performance in assisting endoscopists in their choice of stone removal procedure.

## Introduction

Choledocholithiasis is one of the most common gastrointestinal diseases, occurring in approximately 10% to 20% of gallstone patients [1, 2]. Since the introduction of endoscopic biliary sphincterotomy (EST) in 1974 [3, 4], endoscopic biliary sphincterotomy and stone extraction has been widely used as the primary method of treatment for patients with common bile duct (CBD) stones, with an 80–96% success rate [5, 6]. However, bile duct stone removal with standard biliary sphincterotomy plus stone extraction has a 15% failure rate [7].

Several factors contribute to the difficulty of removing common bile duct stones, such as large stones with diameters of 10 or 15 millimeters, multiple stones, square-shaped stones, distal narrowing, short distal arm CBD, and the ratio of stone to common bile duct diameter greater than one, which makes extraction difficult.

Endoscopic papillary balloon dilation (EPBD) was introduced in 1982 as an alternative for EST [8]. It is performed by dilation of the biliary sphincter with a balloon less than 10 millimeters in diameter; however, it is less effective than EST in removing bile duct stones due to the limited diameter of the orifice dilation with EPBD [9].

Endoscopic papillary large balloon dilation (EPLBD) was introduced in 2003 to assist the removal of large or difficult bile duct stones. It utilizes balloons with a diameter of 12 to 20 millimeters and is used in combination with EST [10].

The European Society of Gastrointestinal Endoscopy (ESGE) and the American Society for Gastrointestinal Endoscopy (ASGE) recently released guidelines for difficult CBD stones, a combination of EST and EPLBD is recommended as a first-line treatment [11, 12].

In developing countries, the facilities and equipment available to endoscopists in primary and secondary hospitals are limited. Within the realm of endoscopic CBD stone treatment, the only options available are standard biliary sphincterotomy and stone extraction. The chances of removing difficult stones are unlikely using standard biliary sphincterotomy plus stone extraction methods during ERCP. Consequently, it may be appropriate to initiate with a limited EST and EPLBD. However, if unsuccessful, difficult stones, require additional procedures and interventions such as mechanical lithotripsy, cholangioscopy-assisted electro-hydraulic/laser lithotripsy, or ESWL. In that case, the patient must either plan the appropriate intervention or be referred to a center that provides that service. However, there is no universal agreement on the degree of difficulty associated with endoscopic stone removal, which is critical in determining the technical outcome of endoscopic CBD stone extractions. Therefore, this study aims to develop a clinical prediction rule for predicting the difficulty of CBD stone removal to assist in decision-making for endoscopists working in resource limited settings in planning appropriate interventions or making referrals to other centers that could perform them.

## Materials and methods

### Source of data

This retrospective cohort study was carried out at Pattani Hospital, which is located in the deep south of Thailand, between January 2017 and June 2022. The study population included all consecutive patients with newly diagnosed CBD stones that underwent ERCP for stone clearance regardless of stone removal difficulty. The Institutional Review Board of Pattani Hospital, Thailand, reviewed and approved this study, which was assigned the number 2020-07-013. Because this study is based on retrospective data, the Institutional Review Board did not require patient consent to review their medical records. Therefore, no individual-level data were utilized, and all data were maintained confidentially following the Helsinki Declaration.

### Participants

Data was collected from all adult patients (age >18 years old) who had a CBD stone on cholangiography and underwent endoscopic treatment at the Pattani secondary hospital. In this study, experienced endoscopists performed over 300 ERCP procedures. For patients that underwent endoscopic CBD stone treatment with standard biliary sphincterotomy and stone extraction or additional procedures; mechanical lithotripsy or EPLBD were available. The standard treatment for all patients diagnosed with a CBD stone was endoscopic sphincterotomy followed by stone extraction. If the stone is successfully removed, refer to complete treatment. If the stone was unsuccessfully removed by standard endoscopic treatment, EPLBD was attempted, followed by CBD stone removal using a basket or balloon catheter, or mechanical lithotripsy was tried. When stones could not be extracted entirely during the first endoscopic procedure, a temporary plastic stent was inserted to prevent biliary obstruction, and subsequent endoscopic procedures were repeated until complete removal was achieved. Patients with an undetected CBD stone on cholangiography, a distal CBD stricture, periampullary or biliary tract cancer, an intrahepatic duct stone, and unsuccessful stone removal by endoscopic treatment were excluded. All patients in this study were performed with a side-viewing Olympus duodenoscope (Olympus, TJF-Q180V), and a standard pull-type sphincterotome was used for all ERCP, EST, and stone extraction procedures under general IV sedation. In addition, cholangiography was performed using a balloon occluded catheter to diagnose and confirm complete stone clearance.

### Outcomes

In this study, patients whose CBD stones have been successfully removed via sphincterotomy and stone extraction are referred to as non-difficult CBD stones. In contrast, patients with CBD stones that require EPLBD in addition to a standard procedure during the initial or subsequent therapy sessions are referred to as difficult CBD stones. These two groups are recognized during the procedure and after complete treatment.

### Predictors

This study documented and categorized clinical parameters into four groups: general characteristics, intraoperative data, endoscopic procedure, and adverse events. 1) General characteristics include age, gender, body mass index (BMI), underlying disease, acute cholangitis, and previous cholecystectomy. 2)Intraoperative information includes endoscopic findings such as: periampulla diverticulum, cholangiographic findings such as: mid-CBD diameter and, distal CBD diameter that was measured at 1 cm proximal to the main ampulla's orifice [13]. The CBD angle is defined as the angle formed by the CBD between 1 cm below the bifurcation and

1 cm above the papilla [14–16]. The distal CBD arm is defined as the distance between the ampullary orifice to the distal CBD angulation point, the longitudinal diameter and transverse diameter of stone at the point of maximum diameter. The difference of stone to distal CBD diameter was calculated by CBD stone diameter (mm) minus the distal CBD diameter (mm). The CBD stone and CBD diameter were corrected for radiograph magnification by multiplying the measured diameter of CBD or stone with the ratio of the actual and measured diameters of the duodenoscope on cholangiography [17]. The stone/CBD ratio, the number of stones, the shape of stone was categorized into oval, round, squared, or fragmented shape, and impacted stone. 3) Endoscopic procedures include endoscopic sphincterotomy (EST), stone extraction by balloon and/or basket, endoscopic papillary balloon dilation (EPBD), mechanical lithotripsy (ML), and stent use. 4) Adverse events include post-ERCP pancreatitis and bleeding in 24 hours after the procedure.

In this study, a model was developed to predict the technical difficulty of endoscopic CBD stone extractions. Therefore, the candidate predictive factors used in the prediction model were selected based on factors associated with CBD stone and CBD's anatomy that is known to influence technical difficulty.

## Sample size

The estimated sample size was based on the standard suggestion of 10 occurrences of interest for each predictor [18]. Therefore, about 70 participants with CBD stones requiring EPLBD in addition to conventional treatment were expected to be used in a model with no more than seven predictors.

## Missing data

The study included all consecutive patients who underwent ERCP for stone clearance. The related parameters were preselected and documented, and a complete-case analysis was performed with no missing data.

## Statistical analysis methods

STATA (version 17.0, Stata Corp LLC, College Station, TX, USA) was used for all statistical calculations. Continuous data with a normal distribution were presented with means and standard deviations (SD). Non-normally distributed data was expressed in median and interquartile range (IQR). The continuous data were evaluated using the t-test or the Mann–Whitney U test, as appropriate. Categorical data were described by frequency and percentage and tested with Fisher's exact probability test. Statistical significance was defined as a p-value less than 0.05.

All parameters underwent exploratory analysis using univariable logistic regression. The odds ratio (OR) its p-value, and the area under the receiver operating characteristics (AuROC) in addition to the 95% confidence interval were investigated separately for each variable. Then, a multivariable logistic regression analysis was carried out to find the EPLBD requirement predictor. The elimination of non-contributing predictors was based on the associated influence of technical difficulty CBD stone removal and statistical significance. First, variables not associated with technical difficulties in CBD stone removals, such as gender and age, were removed. Next, the variables that were multicollinearity were selected and removed. Finally, in the logistic regression model, the variables with an odds ratio closest to 1.00 or an insignificant p-value > 0.1 were sequentially eliminated backward. The insignificant statistical parameters that were known to influence the technical difficulty of CBD stone removal were kept in the model as appropriate.

In the reduced model, a total of four predictors were used. A score was provided for each final predictor based on the logistic regression coefficient of each item. For predicted applicability, the logistic coefficient of each predictor was divided by the model's lowest coefficient and rounded to the nearest non-decimal integer. The model achieves a total score ranging from 0 to 16 with a cut-off of 5 to achieve high specificity for suggesting EPLBD in addition to standard procedures. Positive predictive value (PPV) and likelihood ratio of positive value were computed for each score category to determine the patients' average risk.

The model's predictive performance was evaluated in terms of discrimination and calibration. First, the receiver operating characteristic area (AuROC) was reported as a measure of discrimination. The calibration was reported using Hosmer-Lemeshow goodness-of-fit statistics and a calibration plot comparing the score-predicted EPLBD requirement to the observed EPLBD requirement. Finally, the bootstrap method with 500 replicates was used for internal validation.

## Results

### Participants

During the study period, 200 patients diagnosed with CBD stones underwent endoscopic treatment were collected. The final analysis included 155 patients, as illustrated in Fig 1. There were 111 (71.61%) females and 44 (28.39%) males. Their median age was 52 years (IQR = 37.0–70.0). Additionally, 34 (21.94%) participants had periampulla diverticulum, 22 (14.19%) participants were previous cholecystectomies, and 58 (37.42%) participants had acute cholangitis at the first ERCP session.

The participants were divided into two groups: 76 (49.03%) patients were in the non-difficult group defined as having the successful stone extraction by endoscopic sphincterotomy plus stone extraction and 79 (50.97%) patients were in the difficult group defined as having successful stone extraction by additional EPLBD combined with EST plus stone extraction. The success rate for the non-difficult group in the first session was 73.68%, whereas the success rate for the difficult group was 44.30%. In the non-difficult group, the median of overall session attempts until success is 1 session (IQR = 1.0, 2.0), and in the difficult group, it is 2 sessions (IQR = 1.0, 2.0).

The general characteristics and peri-procedural findings are presented in Table 1. The parameters that found statistical significance on a univariate analysis for the required EPLBD were female, age, mid-CBD diameter, longitudinal diameter of stone, the transverse diameter of stone, and the difference between stone and distal CBD.

### Model development

The analysis used multivariable logistic regression. As described in the method, parameters were selected for the model. Table 2 shows the coefficient, odds ratio (OR), and 95% confidence interval. Two statistically significant predictors, the transverse diameter of the stone and the difference between the stone and distal CBD > 2 mm, and two relevant with technical difficulty but non-statistically significant predictors, stone shape and distal CBD arm length, remained in the final logistic model.

### Score transformation

Each predictor's regression coefficient was used as a weight for score transformation, and the results are presented in Table 3. As demonstrated in Table 3, the scores ranged from 0 to 16, classifying patients into two difficult groups at a cutoff of 5 points. The score for the non-

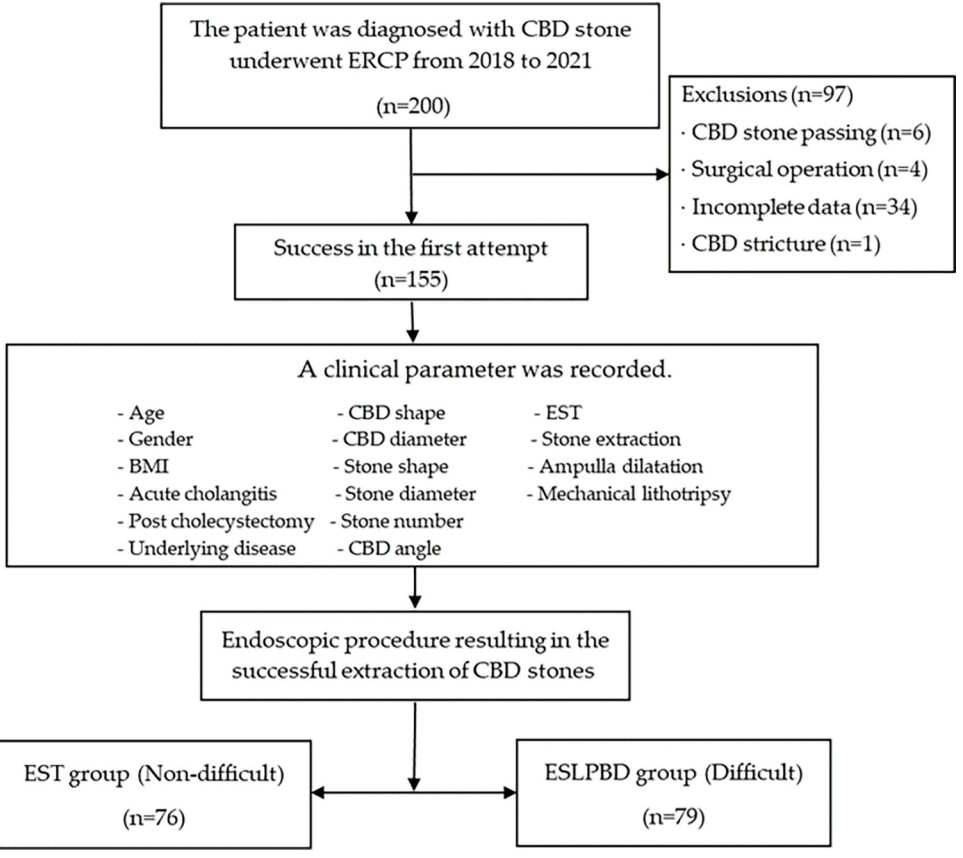

**Fig 1. The study flow diagram.** CBD, common bile duct; ERCP, endoscopic retrograde cholangiopancreatography; EST, endoscopic sphincterotomy; EPLBD, endoscopic large papillary balloon dilatation; ESPLBD, endoscopic sphincterotomy combined with endoscopic papillary large balloon dilatation.

difficult group was 0 to 4 for low-requirement EPLBD, whereas the score for the difficult group was 5 to 16 for high-requirement EPLBD.

## Model performance

The model discrimination performance is demonstrated by the ROC curve of the Difficult CBD stone score (DCSS) model, shown in Fig 2. The area under the receiver operating characteristic curve (AuROC) of the DCSS model revealed an affinity for prediction of 88.39% (95% CI: 0.83, 0.93).

The Hosmer-Lemeshow goodness-of-fit test for the DCSS model also revealed insignificant results (p-value = 0.625). In this data set, the DCSS model proved suitable for predicting EPLBD requirements.

A calibration plot revealed that the score predicted the probability of EPLBD requirement, and the observed risk of EPLBD requirement in the derivation cohort concomitantly increased as the score increased (Fig 3). We selected a cut point value at 5 because of high specificity (94.74%) and sensitivity (72.15%), allowing for the inclusion of individuals with more difficult CBD stones. A score of 5 or higher will likely require EPLBD or further procedures. It was suggested that patients should be referred to a hospital with the necessary equipment to dilate CBD or to make the stones smaller. (Fig 4).

**Table 1. Clinical characteristics, endoscopic finding, cholangiographic finding, procedure and adverse event of difficult CBD stone vs non-difficult CBD stone, evidence of difference (p-value).**

| Clinical Characteristics | Difficult (ESPLBD) (n = 79) | | Non-difficult (EST) (n = 76) | | P-value |
|---|---|---|---|---|---|
| | mean | ±SD | mean | ±SD | |
| **General Characteristics** | | | | | |
| Female (n%) | 63 | 79.75 | 48 | 63.16 | 0.032 |
| Age (years)[a] | 59 | (40.0,75.0) | 48.5 | (35.5,61.5) | 0.010[b] |
| BMI (kg/m$^2$) | 23.98 | ± 4.40 | 23.85 | ± 4.41 | 0.851 |
| Underlying disease(n%) | 21 | 26.58 | 28 | 36.84 | 0.226 |
| Acute cholangitis (n%) | 31 | 39.24 | 27 | 35.53 | 0.740 |
| Post Cholecystectomy (n%) | 8 | 10.13 | 14 | 18.42 | 0.170 |
| **Endoscopic finding** | | | | | |
| Ampulla diverticulum (n%) | 22 | 27.85 | 12 | 15.79 | 0.082 |
| **CBD** | | | | | |
| Mid CBD diameter (mm) | 18.29 | ±5.14 | 13.73 | ± 4.05 | <0.001 |
| Distal CBD diameter (mm) | 10.46 | ±4.45 | 9.73 | ± 3.25 | 0.246 |
| CBD angle (degree) | 148.42 | ±16.64 | 145.91 | ±13.94 | 0.310 |
| Distal CBD arm length (mm) | 37.25 | ±10.23 | 38.32 | ±11.86 | 0.552 |
| **CBD stone** | | | | | |
| Number of stones >3 (n%) | 12 | 15.19 | 8 | 10.53 | 0.475 |
| Longitudinal diameter (mm) | 16.74 | ±8.02 | 9.29 | ±4.21 | <0.001 |
| Transverse diameter (mm) | 13.36 | ±4.54 | 8.10 | ±3.41 | <0.001 |
| Difference of stone to distal CBD diameter (mm) | 2.89 | ±4.13 | -1.63 | ±2.79 | <0.001 |
| **Shape of CBD stone** | | | | | |
| Oval (n%) | 32 | 40.51 | 22 | 28.95 | 0.063 |
| Round (n%) | 23 | 29.11 | 25 | 32.89 | |
| Square (n%) | 13 | 16.46 | 7 | 9.21 | |
| Fragmented (n%) | 11 | 13.92 | 22 | 28.95 | |
| Impacted stone (n%) | 0 | 0 | 5 | 6.58 | |
| **ERCP procedure** | | | | | |
| EST (n%) | 77 | 97.47 | 73 | 96.05 | 0.677 |
| ML (n%) | 5 | 6.33 | 0 | 0 | 0.059 |
| The plastic stent used (n%) | 54 | 68.35 | 58 | 76.32 | 0.287 |
| ERCP session Median[a] | 2 | (1,5) | 1 | (1,2) | |
| **Adverse events** | | | | | |
| Acute pancreatitis (n%) | 15 | 18.99 | 7 | 9.21 | 0.107 |
| Bleeding (n%) | 7 | 8.86 | 1 | 1.32 | 0.064 |

ESPLBD, endoscopic sphincterotomy combined with endoscopic papillary large balloon dilatation; EST, endoscopic sphincterotomy; BMI, body mass index; CBD, common bile duct; ML, mechanical lithotripsy; ERCP, endoscopic retrograde cholangiopancreatography.

[a] Median (Interquartile range)

[b] P-value from rank-sum test

Internal validation was performed by the bootstrap sampling method with 500 replications, which showed constant AuROC 0.884 (95%CI 0.830, 0.938) with model optimism adjusted at AuROC 0.883 (95%CI 0.833, 0.942). The analysis showed acceptable predictive performance.

The ROC curve and AuROC of the DCSS model were compared using the added value concept to an intelligent difficulty scoring and assistance system (DSAS) for endoscopic treatment of common bile duct (CBD) stones that was proposed by Huang et al. [13] (Fig 5.) The DCSS

**Table 2. Coefficient of regression, odds ratio (OR), and 95% confidence interval of the selected parameters and the assignment score of predictive parameters for EPLBD requirements derived from logistic regression after reduced model.**

| Predictors | Coefficients | Odds ratio | 95%CI | p-value | Transformed coefficients | Score |
|---|---|---|---|---|---|---|
| **Transverse diameter of largest stone(mm)** | | | | | | |
| >15 | 3.34 | 28.36 | 4.60,174.70 | <0.001 | 6.60 | 6 |
| 10–15 | 1.13 | 3.09 | 1.13,8.41 | 0.027 | 2.23 | 2 |
| <10 | | 1 | ref | | | 0 |
| **Difference Stone to distal CBD (mm) >2mm** | | | | | | |
| Yes | 3.62 | 37.24 | 7.59, 182.59 | <0.001 | 7.14 | 7 |
| No | | 1 | ref | | | 0 |
| **Stone shape** | | | | | | |
| Square | 1.17 | 3.21 | 0.71, 14.45 | 0.128 | 2.30 | 2 |
| Fragment | 0.57 | 1.77 | 0.44,7.00 | 0.418 | 1.12 | 1 |
| Round | 0.58 | 1.78 | 0.51,4.15 | 0.362 | 1.14 | 1 |
| Oval | | 1 | ref | | | 0 |
| **Distal CBD arm length <36 mm** | | | | | | |
| Yes | 0.51 | 1.66 | 0.66, 4.19 | 0.278 | 1 | 1 |
| No | | 1 | ref | | | 0 |

95% CI, 95% confidence interval; CBD, common bile duct

model's ability to predict the required EPLBD was 88.78%, which increased from the ability to predict the required EPLBD of DSAS was 46.48% (p <0.001).

## Discussion

### Main findings

This study demonstrates that a simple tool, based on stone and CBD-related parameters, was able to stratify the risk of difficult CBD stone endoscopic removal. In particular, the DCSS score was acceptable for evaluating additional procedures required among CBD stone patients who underwent endoscopic treatment. Our final model selected four predictors: stone diameter, the difference between stone and distal CBD diameter, stone shape, and distal CBD arm length.

Many publications report stones larger than 15 millimeters in diameter would predict the probability of unsuccessful stone removal [14–16]. While stones less than 10 millimeters decrease the probability of a failed stone removal. Hence, the size of the stone has been categorized as less than 10 mm, 10–15 mm, and more than 15 mm. This predictor is a strong indication for the use of EPLBD, which indicates difficult CBD stone removal. The larger transverse

**Table 3. Distribution risk of difficult CBD stone clearance, PPV, and 95% CI of PPV.**

| Difficult categories | Score | ESPLBD (n = 79) | | EST (n = 76) | | PPV (95% CI) | LR+ (95% CI) | p-value |
|---|---|---|---|---|---|---|---|---|
| | | n | % | n | % | | | |
| Non-difficult | <5 | 22 | 27.85 | 72 | 94.74 | 23.40(15.29,33.26) | 0.293(0.20,0.42) | <0.001 |
| Difficult | 5–16 | 57 | 72.15 | 4 | 5.26 | 93.44(84.05,98.18) | 13.71(5.23,35.93) | <0.001 |
| Mean ±SE | | 8.32 | ±0.49 | 2.06 | ±0.20 | | | <0.001 |

ESPLBD, endoscopic sphincterotomy combined with endoscopic papillary large balloon dilatation; EST, endoscopic sphincterotomy; PPV, positive predictive value; LR +, likelihood ratio of positive; 95% CI, 95% confidence interval

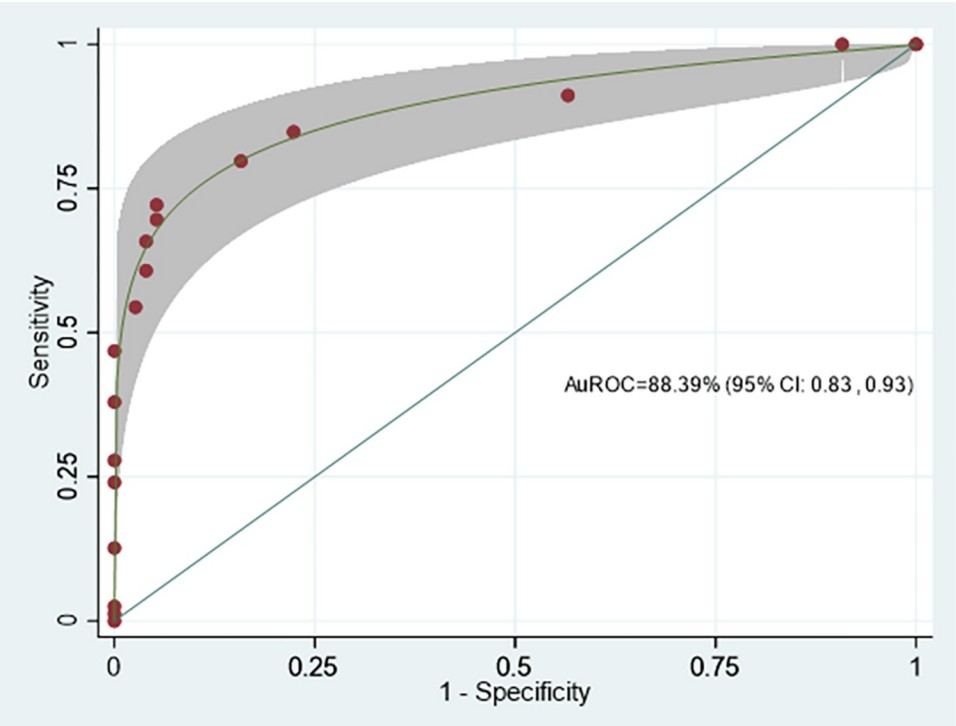

**Fig 2. Performance in discrimination of the clinical model score.** Area Under the Receiver Operating Characteristics (AuROC) curve, and 95% confidence interval band.

diameter stone makes it difficult to remove. Because it may be difficult to grasp while applying basket retraction or passing the stone through a small orifice during swiped balloon, therefore, removing the large stone may require using EPLBD or its fragmentation before extraction.

The difference in diameter between the CBD stone and the distal bile duct, which exceeds 2 mm, was a strong predictor for required EPLBD. We categorized the difference in diameter between the CBD stone and the distal bile duct into two groups based on Sharma et al., implying that stones wider than the distal CBD diameter by more than 2 mm, regardless of stone size, may require an additional stone removal procedure [19]. Furthermore, and the cut-off point of the difference in diameter, which has a high specificity to discriminate between non-difficult CBD stones and the difficult groups in our data, hence the cut-off point of the difference in diameter at two milli-meters was reasonably considered as our predictor.

Despite the fact that stone shape is not a statistically significant predictor in multivariable analyses, this variable was chosen for the prediction model since stone shape influences the difficulty of stone removal in clinical practice. The square form has a greater effect on the EPLBD requirement than other shapes, and the oval form is used as a reference because it is more common than other forms. Therefore, it is logical to assume that because square stones are angular and square, they require more EPLBD than other shapes. So, when a balloon passes over the stone, it gets stuck at the orifice, making it harder to relax the sphincter than with curved or other non-square stones. Additionally, square-shaped stones are difficult to catch in a basket. Therefore, square stones are the ones that require EPLBD more than any other.

A distal CBD arm length shorter than 36 mm was a variable that predicted the difficulty of CBD stone extraction; endoscopists frequently have technical difficulties with CBD stone extraction when the distal CBD arm length is short, just above the major papilla, according to

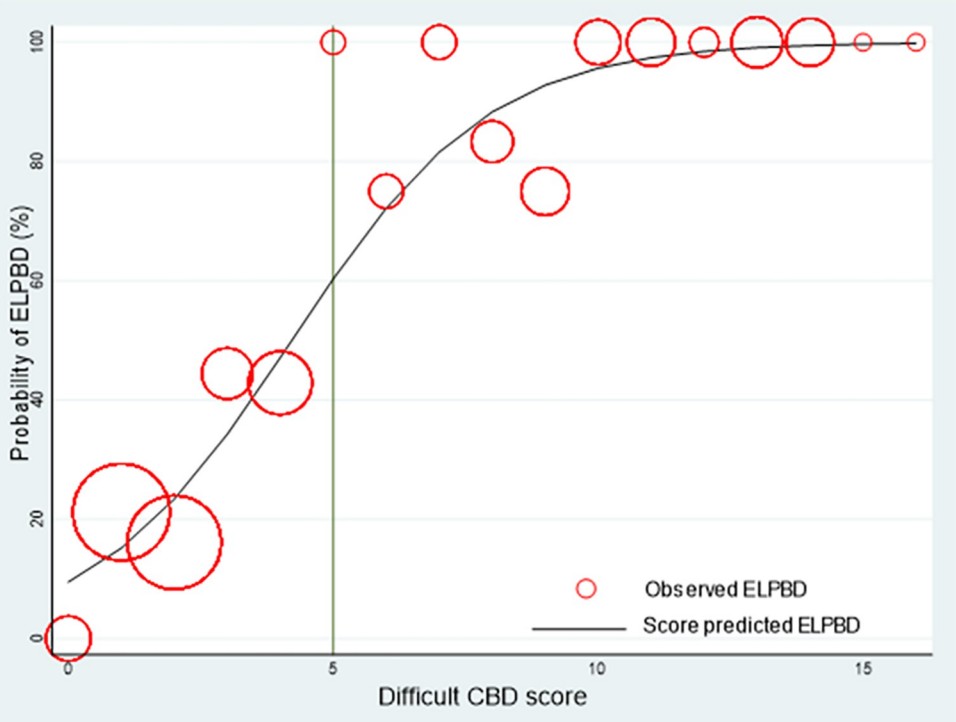

**Fig 3. Calibration plot of score predicted risk versus the observed of requiring EPLBD.** Predicted risk (solid line) versus the observed of requiring EPLBD (circle).

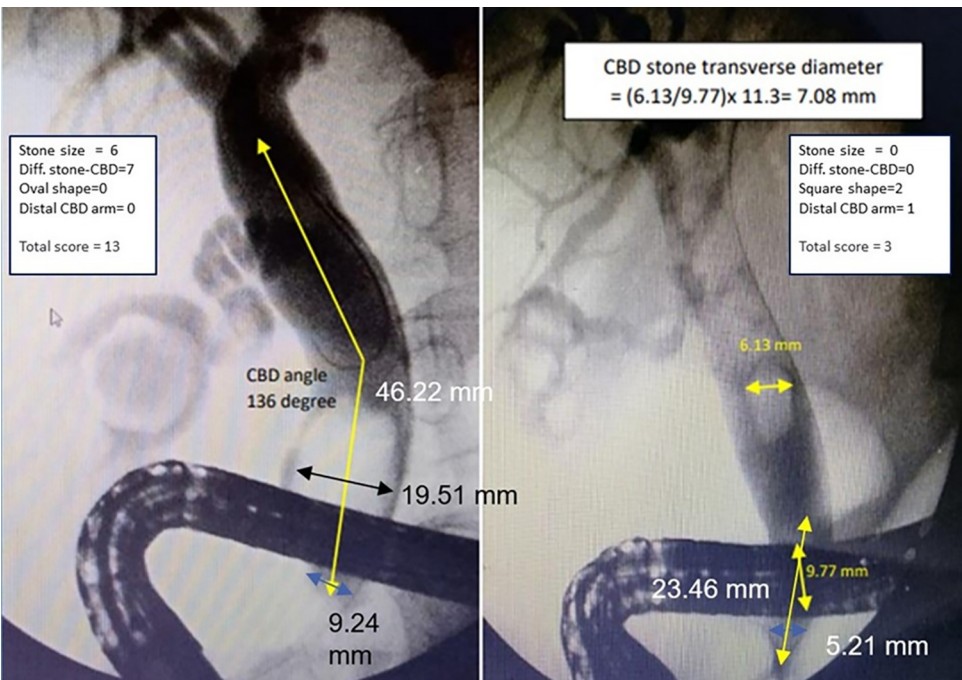

**Fig 4. Cholangiography and measurement of stone and distal CBD diameter.** Total score: 13 (Left) difficult CBD stones should be referred to a hospital with available equipment. Total score: 3 (Right) For non-difficult CBD stones, endoscopic sphincterotomy and stone removal can be used.

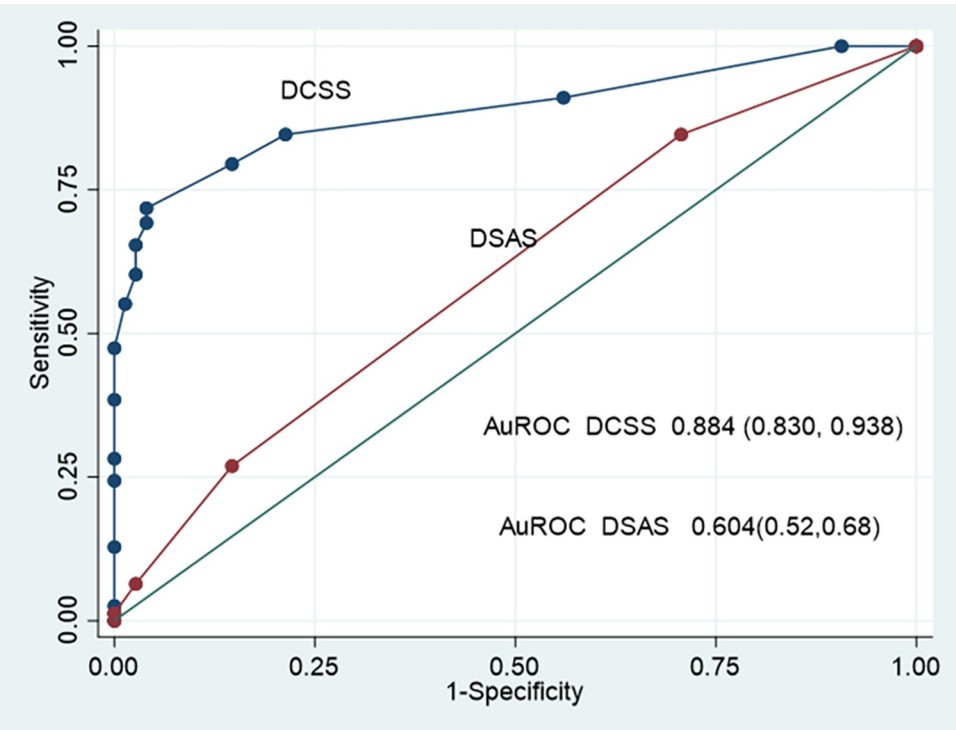

**Fig 5. Comparison between the area under the receiver operating characteristic curve of DCSS and DSAS.** The Difficult CBD stone score (DCSS) model (blue line) and the difficulty scoring and assistance system (DSAS) for endoscopic treatment of common bile duct (CBD) stones (red line).

Kim [20]. This difficulty might be caused by improper positioning and inadequate basket spreading caused by the CBD's anatomic condition. Although insignificant in the multivariable analysis, including this factor in the model improves the prediction performance.

## Comparison with the previous study

Previously, a few clinical prediction rules were proposed to predict the technical difficulty of endoscopic CBD stone removal. In 2021, Huang et al. created an intelligent difficulty scoring and assistance system (DSAS) for the endoscopic treatment of stones in the common bile duct (CBD) [13]. The scoring system included stone and CBD-related factors: stone size, stone number, distal CBD arm length, distal CBD angle, and tapering distal CBD. DSAS uses a stone size of more than 30 mm to predict the difficulty of CBD stone removal. Comparatively, our model categorizes stone sizes as less than 10 mm, 10–15 mm, and larger than 15 mm for the reasons already stated.

The DSAS considered a distal bile duct diameter of less than 6 mm to be a predictor of difficult stones, regardless of the relationship between the size of the stone and the diameter of the distal CBD. In contrast, for reason mentioned earlier, our model considers the disparity between the two diameters a strong predictor.

Although the number of stones [21] and distal CBD angle [20] were proposed factors contributing to unsuccessful of endoscopic stone clearance in patients with difficult CBD stones, both variables were used in DSAS. In our model, we eliminated these two variables because of non-contributing predictors associated with required EPLBD on CBD stone removal in multivariable analysis.

The DSAS was derived from the consensus of five expert endoscopists in order to develop a scoring system based entirely on cholangiography findings, without considering the relevant surgical parameters. However, the DSAS did not describe the model development process and the relationship between each predictor and outcome. In addition, DSAS did not provide model performance; neither discrimination nor calibration was reported, hence restricting the possibility of auditable tools.

### Implication for clinical practice

By using a DCSS score during endoscopic treatment of CBD stones at a hospital with limited resources, the endoscopist can figure out which patients will probably require EPLBD or other procedures. Patients with difficult CBD stones should be referred to a hospital with available equipment. On the other hand, when the probability of EPLBD is low, endoscopic sphincterotomy and stone removal can be used.

### Strengths and limitations

The DCSS was developed to predict the technical difficulty of CBD stone removal based on the association between the predictors and the required EPLBD during endoscopic CBD stone removal. The prediction ability of the DCSS model for the probability of EPLBD was higher compared to the DSAS.

Due to the observational and retrospective nature of this study, data collection and control variables were limited. For instance, the data did not include the number of extraction attempts during the procedure or the stone's consistency, whether rigid or fragile, both of which may have contributed to unsuccessful stone removal. Additionally, stone and CBD diameter measurements were taken in just two dimensions during ERCP; this may have led to an underestimation of the stone's size, shape, and CBD diameter, which would have been more accurate if measured in three dimensions. Future research should study prospective data collection and large sample sizes to improve stone and bile duct assessments.

### Conclusions

This study showed a simple score derived from four stone and CBD-related predictors: stone diameter, the difference between stone and distal CBD diameter, stone shape, and distal CBD arm length. Patients with a DCSS score of 5 or more had a high probability of EPLBD for stone removal. Therefore, it was suggested that patients with difficult CBD stones be referred to a hospital with available equipment. Conversely, in the case of a low probability of EPLBD, endoscopic sphincterotomy and stone extraction can be used for treatment.

### Author Contributions

**Conceptualization:** Chote Wongkanong, Jayanton Patumanond, Thawee Ratanachu-ek, Apichat Tantraworasin.

**Data curation:** Chote Wongkanong.

**Formal analysis:** Chote Wongkanong, Jayanton Patumanond.

**Methodology:** Chote Wongkanong, Jayanton Patumanond, Apichat Tantraworasin.

**Project administration:** Chote Wongkanong, Apichat Tantraworasin.

**Supervision:** Jayanton Patumanond, Thawee Ratanachu-ek, Apichat Tantraworasin.

**Validation:** Sunhawit Junrungsee.

**Visualization:** Sunhawit Junrungsee.

**Writing – original draft:** Chote Wongkanong.

**Writing – review & editing:** Jayanton Patumanond, Thawee Ratanachu-ek, Sunhawit Junrungsee, Apichat Tantraworasin.

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
