## [Decision Letter · Decision Letter 0]

18 Jan 2023

PONE-D-22-32458Development of clinical prediction rule for the requirement of endoscopic large papillary balloon dilation (EPLBD) on endoscopic CBD stone clearance.PLOS ONE

Dear Dr. Tantraworasin,

Thank you for submitting your manuscript to PLOS ONE. After careful consideration, we feel that it has merit but does not fully meet PLOS ONE’s publication criteria as it currently stands. Therefore, we invite you to submit a revised version of the manuscript that addresses the points raised during the review process.

We look forward to receiving your revised manuscript.

Kind regards,

Roberto Coppola, MD, FACS

Academic Editor

PLOS ONE

Journal Requirements:

"This research work was partially supported by Chiang Mai University, Chiang Mai, Thailand."

4. Please include a caption for figure 5.

5. Please ensure that you refer to Figure 5 in your text as, if accepted, production will need this reference to link the reader to the figure.

Additional Editor Comments:

Reviewer 1

In the present study, the authors explore a very interesting idea about developing a predictive score for EPLBD needs during ERCP. The paper is well written. This is an exciting topic for all advanced endoscopists. They found that the size of CBD stones, stone shape, and distal CBD arm length <36 mm could be used to calculate the score that in case of being ≥ 5, the PPV was high. This is an interesting result; however, I have some concerns:

1. The author did not mention if they included only adult patients

2. Were the procedures (ERCPs) done under MAC or general IV anesthesia?

3. Which was the number and experience of the endoscopists?

4. One-third of the patients who required EPLBD have scores <5….this seems to be a very high proportion.

5. Maybe my most important question is the following: this new score that the authors are proposing needs information obtained through/during the ERCP procedure. After you get it you calculate the score (ex. To calculate the CBD angle, you need to measure the angle formed by the CBD between 1 cm below the bifurcation and 1 cm above the papilla, the distance between the ampullary orifice to the distal CBD angulation point, the longitudinal diameter and transverse diameter of the stone at the point of maximum diameter, etc)….so which is the advantage in doing all this (with the time that takes to do it) vs to do a swept with the extraction balloon (and if you see is not possible, go directly to EPLBD or refer the patient).

6. PLOS authors have the option to publish the peer review history of their article (what does this mean?). If published, this will include your full peer review and any attached files.

Reviewer 2

This paper developed a simplified scoring system for clinical prediction of difficulty in CBD stone removal to assist endoscopists working in resource-limited settings in deciding whether to proceed with an intervention or refer patients to a center capable of performing additional procedures and interventions.The paper is very well written, and contributes a scoring system which has acceptable prediction performance in assisting endoscopists in their choice of stone removal procedure.

Reviewers' comments:

Reviewer's Responses to Questions

**Comments to the Author**

1. Is the manuscript technically sound, and do the data support the conclusions?

Reviewer #1: Partly

Reviewer #2: Yes

2. Has the statistical analysis been performed appropriately and rigorously? 

Reviewer #1: Yes

Reviewer #2: Yes

3. Have the authors made all data underlying the findings in their manuscript fully available?

Reviewer #1: Yes

Reviewer #2: Yes

4. Is the manuscript presented in an intelligible fashion and written in standard English?

Reviewer #1: Yes

Reviewer #2: Yes

5. Review Comments to the Author

Reviewer #1: In the present study, the authors explore a very interesting idea about developing a predictive score for EPLBD needs during ERCP. The paper is well written. This is an exciting topic for all advanced endoscopists. They found that the size of CBD stones, stone shape, and distal CBD arm length <36 mm could be used to calculate the score that in case of being ≥ 5, the PPV was high. This is an interesting result; however, I have some concerns:

1. The author did not mention if they included only adult patients

2. Were the procedures (ERCPs) done under MAC or general IV anesthesia?

3. Which was the number and experience of the endoscopists?

4. One-third of the patients who required EPLBD have scores <5….this seems to be a very high proportion.

5. Maybe my most important question is the following: this new score that the authors are proposing needs information obtained through/during the ERCP procedure. After you get it you calculate the score (ex. To calculate the CBD angle, you need to measure the angle formed by the CBD between 1 cm below the bifurcation and 1 cm above the papilla, the distance between the ampullary orifice to the distal CBD angulation point, the longitudinal diameter and transverse diameter of the stone at the point of maximum diameter, etc)….so which is the advantage in doing all this (with the time that takes to do it) vs to do a swept with the extraction balloon (and if you see is not possible, go directly to EPLBD or refer the patient).

Reviewer #2: This paper developed a simplified scoring system for clinical prediction of difficulty in CBD stone removal to assist endoscopists working in resource-limited settings in deciding whether to proceed with an intervention or refer patients to a center capable of performing additional procedures and interventions.The paper is very well written, and contributes a scoring system which has acceptable prediction performance in assisting endoscopists in their choice of stone removal procedure.

6. PLOS authors have the option to publish the peer review history of their article (what does this mean?). If published, this will include your full peer review and any attached files.

Reviewer #1: No

Reviewer #2: No

---

## [Author Response · Author response to Decision Letter 0]

7 Feb 2023

Dear academic Editor

Thank you for giving me the opportunity to submit a revised draft of my manuscript titled 

"Development of clinical prediction rule for the requirement of endoscopic large papillary balloon dilation (EPLBD) on endoscopic CBD stone clearance" to PLOS ONE. I appreciate the time and effort that you and the reviewers have dedicated to providing your valuable feedback on my manuscript. I am grateful to the reviewers for their insightful comments on my paper. I have been able to incorporate changes to reflect most of the suggestions provided by the reviewers. I have highlighted the changes within the manuscript. Here is a point-by-point response to the comments and concerns.

Journal Requirements:

1. Please ensure that your manuscript meets PLOS ONE's style requirements, including those for file naming……….

Reply: We have revised the manuscript to meet the style requirements of PLOS ONE for the main body, title, authors, and affiliations.

"This research work was partially supported by Chiang Mai University, Chiang Mai, Thailand."

Reply: We removed the funding-related text "Chiang Mai University partially supported this research work, Chiang Mai, Thailand," from our manuscript. And we will add the funding in the funding statement.

Reply: The ethics statement is only in the Materials and methods section of our manuscript. So, the ethical information is only written here on page 5, line 4 of the first paragraph.

4. Please include a caption for figure 5.

Reply: The caption for figure 5 appears on page 22 of our manuscript 

"Fig 5. Comparison between the area under the receiver operating characteristic curve of DCSS and DSAS.

The Difficult CBD stone score (DCSS) model (blue line) and the difficulty scoring and assistance system (DSAS) for endoscopic treatment of common bile duct (CBD) stones (red line)".

5. Please ensure that you refer to Figure 5 in your text as, if accepted, production will need this reference to link the reader to the figure.

Reply: Figure 5 was referred in our text on page 12, line 6 of the second paragraph.

Additional Editor Comments:

Response to Reviewers

Reviewer 1

Reviewer: 1.1. The author did not mention if they included only adult patients

Reply: This study included only adult patients. Thank you for pointing this out. I agree with this comment. Therefore, I have clearly described the purpose of this study on page 5, line 1 of paragraph 2.

Reviewer: 1.2. Were the procedures (ERCPs) done under MAC or general IV anesthesia?

Reply: Agree. I have, accordingly, clarified to emphasize this point. All patients were done procedures under general IV sedation. Therefore, I have clearly described this on page 6, line 6 of paragraph 1.

Reviewer: 1.3. Which was the number and experience of the endoscopists?

Reply: In this study, experienced endoscopists performed over 300 ERCP procedures. Therefore, I have described in detail on page 5, line 2 of paragraph 2.

Reviewer: 1.4. One-third of the patients who required EPLBD have scores <5….this seems to be a very high proportion.

Reply: You have raised an important point here. However, We selected a cut point value at 5 because of high specificity (94.74%) and sensitivity (72.15%), allowing for the inclusion of individuals with more difficult CBD stones. In addition, at this cut point, It early realizes that bile duct stones are difficult to treat and need a dilation or to make the stones smaller. Therefore, I have clearly described on page 12, line 3 of paragraph 1.

Reviewer: 1.5. Maybe my most important question is the following: this new score that the authors are proposing needs information obtained through/during the ERCP procedure. After you get it you calculate the score (ex. To calculate the CBD angle, you need to measure the angle formed by the CBD between 1 cm below the bifurcation and 1 cm above the papilla, the distance between the ampullary orifice to the distal CBD angulation point, the longitudinal diameter and transverse diameter of the stone at the point of maximum diameter, etc)….so which is the advantage in doing all this (with the time that takes to do it) vs to do a swept with the extraction balloon (and if you see is not possible, go directly to EPLBD or refer the patient).

Reply: You have raised an important point here. However, in the case of our study, this simple scoring system is easy to use by estimating the diameter of the CBD stone, distal CBD diameter, and distal CBD arm compared to the actual diameter of the duodenoscope by the rule of three calculators based on proportion with the short time to do it during ERCP. The stone shape was observed directly on cholangiography. Patients who scored more than 5 indicated that it was necessary to gain CBD access or reduce the size of the stone before stone clearance, which aids in determining the following procedure and choice of sedation because it prolongs operations and aids in deciding whether to proceed with the procedure or refer the patient based on feasibility in that hospital.

Reviewer: 1.6. PLOS authors have the option to publish the peer review history of their article (what does this mean?). If published, this will include your full peer review and any attached files.

Reply: We follow the guideline for publication of PLOS one. If accepted for publication, published version of manuscript will be followed with PLOS guideline.

Reviewer 2

This paper developed a simplified scoring system for clinical prediction of difficulty in CBD stone removal to assist endoscopists working in resource-limited settings in deciding whether to proceed with an intervention or refer patients to a center capable of performing additional procedures and interventions.The paper is very well written,and contributes a scoring system which has acceptable prediction performance in assisting endoscopists in their choice of stone removal procedure.

Reply: Thank you very much for your comment. 

We look forward to hearing from you in due time regarding our submission and to responding to 

any further questions and comments you may have.

Sincerely,

Chote Wongkanong M.D. 

Apichat Tantraworasin M.D., Ph.D.

---

## [Editor Report · Decision Letter 1]

27 Feb 2023

Development of clinical prediction rule for the requirement of endoscopic papillary large balloon dilation (EPLBD) on endoscopic CBD stone clearance.

PONE-D-22-32458R1

Dear Dr. Tantraworasin,

We’re pleased to inform you that your manuscript has been judged scientifically suitable for publication and will be formally accepted for publication once it meets all outstanding technical requirements.

Kind regards,

Roberto Coppola, MD, FACS

Academic Editor

PLOS ONE

Additional Editor Comments (optional):

The authors thoroughly answered the Reviewer's requests. I have decided to accept the paper for publication.
---

## [Editor Report · Acceptance letter]

6 Mar 2023

PONE-D-22-32458R1 

Development of clinical prediction rule for the requirement of endoscopic papillary large balloon dilation (EPLBD) on endoscopic CBD stone clearance. 

Dear Dr. Tantraworasin:

I'm pleased to inform you that your manuscript has been deemed suitable for publication in PLOS ONE. Congratulations! Your manuscript is now with our production department. 

Kind regards, 

on behalf of

Professor Roberto Coppola 

Academic Editor

PLOS ONE